# Pyk2/FAK Signaling Is Upregulated in Recurrent Glioblastoma Tumors in a C57BL/6/GL261 Glioma Implantation Model

**DOI:** 10.3390/ijms241713467

**Published:** 2023-08-30

**Authors:** Jescelica Ortiz Rivera, Grace Velez Crespo, Mikhail Inyushin, Yuriy Kucheryavykh, Lilia Kucheryavykh

**Affiliations:** Department of Biochemistry, School of Medicine, Universidad Central de Caribe, Bayamon, PR 00956, USA; 416gvelez@uccaribe.edu (G.V.C.); mikhail.inyushin@uccaribe.edu (M.I.); yuriy.kucheryavykh@uccaribe.edu (Y.K.); lilia.kucheryavykh@uccaribe.edu (L.K.)

**Keywords:** glioblastoma, recurrent tumor, Pyk2, FAK

## Abstract

The majority of glioblastomas (GBMs) recur shortly after tumor resection and recurrent tumors differ significantly from newly diagnosed GBMs, phenotypically and genetically. In this study, using a Gl261-C57Bl/6 mouse glioma implantation model, we identified significant upregulation of proline-rich tyrosine kinase Pyk2 and focal adhesion kinase (FAK) phosphorylation levels—pPyk2 (579/580) and pFAK (925)—without significant modifications in total Pyk2 and FAK protein expression in tumors regrown after surgical resection, compared with primary implanted tumors. Previously, we demonstrated that Pyk2 and FAK are involved in the regulation of tumor cell invasion and proliferation and are associated with reduced overall survival. We hypothesized that the use of inhibitors of Pyk2/FAK in the postsurgical period may reduce the growth of recurrent tumors. Using Western blot analysis and confocal immunofluorescence approaches, we demonstrated upregulation of Cyclin D1 and the Ki67 proliferation index in tumors regrown after resection, compared with primary implanted tumors. Treatment with Pyk2/FAK inhibitor PF-562271, administered through oral gavage at 50 mg/kg daily for two weeks beginning 2 days before tumor resection, reversed Pyk2/FAK signaling upregulation in recurrent tumors, reduced tumor volume, and increased animal survival. In conclusion, the use of Pyk2/FAK inhibitors can contribute to a delay in GBM tumor regrowth after surgical resection.

## 1. Introduction

Glioblastoma (GBM) is the most common and aggressive type of adult brain tumor. Tumor resection is the first-line therapy for GBM; however, the invasive nature of GBM cells reduces the efficacy of surgical resection, as single glioma cells that are left after surgical resection in the tumor surrounding brain parenchyma soon lead to tumor recurrence, and a majority of GBMs recur shortly after resection [1,2,3,4]. Less than two percent of GBM patients survive five years after diagnosis, mostly due to tumor relapse within six to nine months of resection, and the average length of survival is estimated to be 12 to 15 months [5,6].

The faster proliferation index and the resistance to treatment of recurrent GBMs have been correlated with molecular and signaling differences between newly diagnosed and recurrent tumors [7]. Multiple mutations that are not present in primary glioblastomas have been detected in recurrent GBMs, raising the possibility that the residual tumor cells that survive the initial treatment of tumor resection and concurrent chemo- and radiotherapy may acquire new mutations that favor increased treatment resistance, invasion, migration, and adaptation to the new microenvironment [8]. Based on gene-expression analysis of paired GBM specimens from the same patients at initial diagnosis and after tumor recurrence, it was shown that primary GBMs tend to shift toward a more mesenchymal profile upon recurrence, which is associated with more aggressive behavior [9,10]. Analyses of gene-expression patterns in primary and recurrent GBMs have shown that tumors with a non-mesenchymal gene-expression signature transition to a more mesenchymal signature upon recurrence. This transition is often accompanied by amplification or mutations in the epidermal growth factor receptor (EGFR), loss of heterozygosity at neurofibromin 1 (NF1), and an increase in isocitrate dehydrogenase 1 (IDH1) mutations. Conversely, tumors with a mesenchymal gene-expression signature typically exhibit greater stability and only occasionally deviate from their mesenchymal signature [6,9,10,11,12]. Reductions in tumor vascularity and the CD34 expression marker of endothelial cells have also been shown in recurrent GBM tumors, compared with primary tumors [10]. Additionally, deregulation of the dysregulated cell cycle, the JAK/STAT pathway, histone methylation, platelet-derived growth factor receptor A (PDGFRA), and telomerase reverse transcriptase (TERT) have been reported to be more frequent in recurrent gliomas, compared with newly diagnosed tumors [6,13,14]. 

Focal adhesion kinase (FAK) and proline-rich tyrosine kinase 2 (Pyk2) are nonreceptor tyrosine kinases that link integrins and plasma membrane receptors to intracellular signaling. They have been shown to be associated with cell proliferation, migration, and survival regulation in glioma cells [15,16,17,18]. Pyk2 and FAK respond to elevated intracellular calcium levels or to the clustering of integrins, respectively, by initiating self-phosphorylation processes. This activation leads to downstream Src signaling, followed by additional phosphorylation at specific sites—Tyr579, Tyr580, and Tyr881 for Pyk2, and Tyr407, Tyr576, Tyr577, Tyr871, and Tyr925 for FAK—and triggering the activation of downstream signaling pathways, including Ras, MAP kinase, and NF-kappaB [19,20,21]. 

Moreover, Pyk2 and FAK signaling can be triggered by a wide range of transmembrane receptors, including G protein-coupled receptors, tyrosine receptors, and cytokine receptors [22,23,24]. Activation of these receptors can be triggered by cytokines and chemokines that are expressed in the tumor microenvironment, particularly by tumor-infiltrating myeloid cells (TIMs), or via the paracrine expression of various cytokines [24]. Additionally, these pathways can be directly activated through phosphorylation by diverse kinases [25,26]. It has been demonstrated that Pyk2 overexpression induces the migration and invasion of tumor cells [27,28], while increased FAK activity has also been correlated with increased cell proliferation and cell-cycle progression [29,30], which could contribute to tumor growth. However, our previous studies, performed on primary human GBM cell cultures developed from tumor specimens and on mouse glioma implantation models, demonstrated that both Pyk2 and FAK are involved in cell-cycle and invasion regulation, and that the extent of this involvement depends on the distinct levels of Pyk2 or FAK expression within each tumor [31,32]—greater expression and activation of either of these kinases results in a more prominent role for that kinase, relative to the other, in governing cell-cycle progression and invasion. 

In this study, we identified that tumors regrown after surgical resection represent an upregulation of Pyk2- and FAK-signaling activation. We hypothesized that the use of small-molecule Pyk2/FAK inhibitors could eliminate the effect of Pyk2/FAK upregulation and reduce the growth of recurrent tumors. A majority of studies that identified genetic and phenotypic differences between newly diagnosed and recurrent GBMs were performed with human tumor specimens. As a majority of GBM patients undergo chemo- and radiotherapeutic treatments after tumor resection surgery, it is not clear whether identified differences between newly diagnosed and recurrent specimens are related directly to signaling and genetic modulations in recurrent tumors or caused by chemo- and/or radiotherapeutic exposure. The use of animal models is a convenient tool for discriminating between the effects of surgical intervention and post-treatment modifications. 

In this study, using a GL261-C57Bl/6 mouse glioma implantation model, we investigated the effect of tumor resection on Pyk2 and FAK phosphorylation levels in tumors that had regrown after surgical resection. This study demonstrated significant upregulation of Pyk2 and FAK phosphorylation in regrown tumors, compared with primary implanted tumors. PF-562271, a small-molecule inhibitor of Pyk2 and FAK that was provided orally 2 days before a planned surgical tumor resection and continued throughout the post-surgical period significantly reduced tumor growth and increased the median survival of the animals by 33%, compared with non-treated controls. 

## 2. Results

### 2.1. Pyk2 and FAK Phosphorylation Is Upregulated in Tumors That Have Regrown after Resection

Western blot analysis was performed to evaluate the levels of expression of total and phosphorylated Pyk2 and FAK in primary implanted and regrown-after-resection tumors in a GL261-C57BL/6 mouse glioma implantation model. Analysis of primary implanted tumors was conducted 14 days after the initial tumor implantation. Additionally, regrown tumors were examined 14 days after the surgical resection of the initially implanted tumors. A 3-fold upregulation of pPyk2 (579/580) and a 2.5-fold upregulation of pFAK (925) were detected in regrown tumors, compared with primary implanted tumors (Figure 1). The total protein expression of Pyk2 and FAK did not show significant changes in either primary or regrown tumors. This correlated with PTK2 (FAK) and PTK2B (Pyk2) gene-expression analysis of 540 GBM cases from the glioblastoma tumor RNA-Sequence dataset (R2 Genomic Analysis Visualization Platform), indicating no significant difference in Pyk2 and FAK gene expression between newly diagnosed and recurrent GBM tumors (Appendix A). Whole Western blot images and loading controls are presented in Appendix A.

### 2.2. Pyk2 and FAK Gene-Expression Pattern in Glioblastoma Tumors Associate with Overall Patient Survival

For the analysis of survival probability, clinical data, and expression values for PTK2 and PTK2B genes (which code for FAK and Pyk2, respectively) in GBM tumors (provisional) were obtained from the cBioPortal for cancer genomics. Kaplan–Meier analysis of 619 glioblastoma cases from the glioblastoma tumor RNA-Sequence dataset identified a negative association between levels of gene expression of PTK2 and PTK2B and overall survival (Figure 2). The median survival probability in a cohort of patients with high gene expression versus low gene expression in tumor specimens was 12.94 months vs. 16.79 months (*p* = 0.009) for PTK2B and 12.58 months vs. 13.63 months (*p* = 0.0275) for PTK2. 

### 2.3. PF-562271 Reduces Pyk2 and FAK Phosphorylation and the Ki67 Proliferation Index in Tumors Regrown after Surgical Resection

To evaluate the extent of pharmacological inhibition of Pyk2/FAK signaling in tumors regrown after surgical resection, Western blot analysis of tumors was performed for the animals that received Pyk2/FAK inhibitor PF-562271 via oral gavage or vehicle beginning 2 days before the tumor resection and, then, up to 14 days after. Five animals per experimental group were used. Analysis of tumors was performed on the 16th day of treatment (Figure 3A). Downregulation of pPyk2 (579/580) by 58% and 50% downregulation of pFAK (925) were detected in the tumors of animals that received PF-562271, compared with control animals that received the vehicle, without significant modifications in total Pyk2 and FAK expression between the experimental groups (Figure 3B). Additionally, considering that tumor-infiltrating microglia and macrophages (TIM) have been demonstrated to rely on Pyk2 and FAK [20], the impact of PF-562271 on the infiltration of these cells into tumors was assessed. The Ionized Calcium-Binding Adapter Molecule 1 (Iba1) was used as a marker for infiltrating myeloid cells, uncovering a 47% decrease in Iba1 expression within tumors treated with PF-562271, compared to the control group. Confocal immunofluorescent analysis further confirmed a significant reduction of tumor-infiltrating myeloid (TIM) cells stained with galactose-binding lectin in the PF-562271-treated group, compared with the control tumors (Figure 3C,D). These data suggest that PF-562271 treatment reversed the upregulation of Pyk2 and FAK signaling in regrown tumors and significantly reduced the TIM infiltration of tumors.

Additionally, PF-562271 treatment significantly affected Cyclin D1 expression in tumors regrown after surgical resection. Western blot analysis revealed 50% downregulation of Cyclin D1 in the tumors of animals that received PF-562271 for 14 consecutive days, compared with control animals that received the vehicle (Figure 4A). Further immunofluorescence evaluation of tumor sections identified a 20% decrease in the number of cells expressing Ki-67 in the regrown tumors in the animals treated with PF-562271, compared with the non-treated control group (Figure 4B,C). Whole Western blot images and loading controls are presented in Appendix A.

### 2.4. PF-562271 Reduces the Growth of Recurrent Tumors and Increases Animal Survival

Analysis of tumor size with the use of hematoxylin and eosin (H&E) staining of brain sections was performed to evaluate the effect of PF-562271 on tumor growth after surgical resection. The animals were administered either the vehicle (control group) or PF-562271 (experimental group) starting 2 days before the tumor resection procedure and continuing for 14 days thereafter. Then, the animals were euthanized, and tumor analysis took place. Each experimental group comprised a total of five animals. As demonstrated in Figure 5A,B, a 43% decrease in tumor volume was detected in animals treated with PF-562271 on day 14 after tumor resection, compared with animals that received a vehicle. No noteworthy contrast in indications of necrosis or bleeding within tumors was evident between the control group and animals treated with PF-562271. Additionally, no adverse effects resulting from PF-562271 treatment that significantly impacted the health of the animals were noted. H&E staining for brain sections encompassing individual recurrent tumors from both the vehicle and PF-562271 treatment groups are presented in Appendix A.

For the survival analysis, treatment with PF-562271 or the vehicle was sustained throughout the entire post-tumor resection period until the endpoint for animal termination. Survival analysis further showed a significant increase in median survival in the PF-562271 treated group, compared with animals that received the vehicle (43 days vs. 29 days after tumor resection), which translated to a 33% increase in median survival in the animals treated with PF-562271 beginning 2 days before tumor resection and during the entire post-surgical period (Figure 5D). Overall, the data indicate that PF-562271 treatment targets recurrent tumor growth, significantly reducing tumor size and increasing the animal survival rate.

## 3. Discussion

An important factor in the poor prognosis of primary GBM is the high recurrence rate [33,34]. Surgical tumor resection remains the main treatment for GBM patients, followed by chemotherapy and radiotherapy. However, the role of surgical intervention and its possible influence on the progression of a regrown tumor has not yet been clearly defined. In this study, we identified activation of Pyk2 and FAK signaling without modulation in the basal expression levels of total Pyk2 and FAK in tumors regrown after surgical resection, compared with primary implanted tumors (Figure 1). Pyk2 and FAK are directly involved in the regulation of glioma cell proliferation and invasion, and they can contribute to the growth of recurrent tumors [35,36] and to overall survival probability, as presented in Figure 2. 

This study identified that PF-562271 treatment significantly decreased the expression of Cyclin D1 in regrown tumors (Figure 4). This correlated with the literature, indicating that inhibition of Pyk2 reduced the activation of Wnt/β-catenin signaling by destabilizing β-catenin, leading to the downregulation of c-Myc and Cyclin D1 [37,38]. Furthermore, our data demonstrated a decrease in the Ki67 proliferating index in regrown tumors treated with PF-562271, suggesting that Pyk2 and FAK signaling has a strong influence on cell-cycle regulation and glioma cell proliferation. Previously, it was shown that PF-562271 treatment resulted in reduced tumor growth and invasion in a number of cancer models, including GBM [39,40]. 

A study by Hu et al. demonstrated a reduction in tumor weight and volume upon treatment with PF-562271 in established osteosarcoma xenograft tumors [41], supporting the notion that FAK and Pyk2 are important modulators of invasive and proliferative properties in tumor cells. Prior investigations conducted in our laboratory using primary human GBM cell lines and the GL261/C57Bl/6 mouse GBM model revealed that while PF-562271 treatment yielded a modest pro-apoptotic and anti-proliferative impact on GBM cells, its effects were considerably amplified when administered in conjunction with the chemotherapeutic drug temozolomide. This combination demonstrated increased apoptosis induction and cell-cycle inhibition, compared with temozolomide monotherapy. Furthermore, PF-562271 effectively curtailed the invasive capabilities of primary human GBM cells and was similarly efficacious in a mouse glioma implantation model [42]. 

The current study demonstrated that PF-562271 treatment resulted in a reduction of tumor volume by 43% in animals with post-resected regrown tumors, while increasing the median survival rate by 33%, compared with the untreated group (Figure 5). Overall, the results obtained from this study demonstrated that Pyk2 and FAK inhibitors are a promising strategy for the treatment of recurrent GBMs, targeting Pyk2- and FAK-dependent tumor growth. 

Recurring tumors develop from cells located at the invasive edge that remain within the brain tissue after surgical removal. Previous research has uncovered significant molecular and genetic variations in GBM cells obtained from the central tumor region and the surrounding margin areas [43]. 

Moreover, primary treatments employed for GBM, such as surgical removal and therapies such as radiation and chemotherapy, contribute to the acidification of the microenvironment, which promotes tumor recurrence and complications [44]. The acidic microenvironment creates favorable conditions for the migration and invasion of cancer cells. Studies have revealed that acidic pH induces the activation of acid-sensing ion channels (ASICs), leading to intracellular increases in calcium (Ca2+) levels that trigger epithelial-mesenchymal transition (EMT) in pancreatic cancer cells [45]. pH alterations have also been associated with the activation of Pyk2, which has been identified as a pH sensor and activator in various types of cancer [46].

Distinct differences were observed in the composition and gene-expression profile of tumor-infiltrating myeloid (TIM) cells between the central core and margin of the tumor [44,47]. These differences resulted in the transformation of tumor cells into a more invasive and proliferative phenotype. Myeloid and T cells were found to be equally abundant in both primary and recurrent tumors, but their spatial arrangement and activation within the tumors varied [48,49]. Newly diagnosed GBM tumors were predominantly infiltrated by TIMs with microglial gene signatures, whereas recurrent tumors exhibited a shift toward TIMs with monocyte gene signatures [50]. The high infiltration of monocyte-derived macrophages has been associated with poor survival outcomes in GBM patients [51]. However, our previous studies using a mouse GBM model did not reveal significant differences in the ratio of CD45^high^/CD45^low^ expression between primary implanted and recurrent tumors, indicating no disparity in peripheral macrophage versus microglia infiltration [47]. We did observe alterations in the cytokine expression profile of purified TIMs from recurrent and primary implanted tumors, primarily due to changes in TIM polarization rather than variations in myeloid cell populations infiltrating the tumors. In primary implanted tumors, a predominant CD206+/CD86− TIM cell population was present, whereas recurrent tumors exhibited a predominantly CD206+/CD86+ population. This shift was accompanied by significant modifications in the cytokine expression profile, including sustained upregulation of IL4, IL5, IL10, IL12, IL17, vascular endothelial growth factor (VEGF), and monocyte chemoattractant protein 1 (MCP1/CCL2) in TIM cells from recurrent tumors, compared with primary implanted tumors [47]. 

Our previous studies reported the involvement of TIM-derived cytokines and growth factors such as EGF, PDGF, stromal-derived factor 1 (SDF1), IL8, and IL6 in the upregulation of Pyk2 and FAK signaling in GBM cells, promoting their migration and proliferation [42]. Furthermore, utilizing a CD11b-HSVTK transgenic mouse model of glioma implantation, we demonstrated that the elimination of microglia/macrophages in implanted tumors resulted in a significant downregulation of pPyk2(579/580) in glioma cells [24]. These findings suggested that the increased expression of pro-tumorigenic cytokines in the tumor microenvironment triggered by surgical resection may contribute to the upregulation of Pyk2/FAK phosphorylation, ultimately promoting the growth of recurrent tumors. Considering the intricate interplay involving cytokine-mediated signaling, myeloid cell infiltration, and Pyk2/FAK activation within tumor cells, the decline in the Iba1 marker and the reduced infiltration of TIM observed in regrown tumors following PF-562271 treatment (Figure 3B–D) signifies that PF-562271 treatment not only hampers Pyk2/FAK signaling in tumor cells directly, but also leads to a reduction in infiltrating myeloid cells within the tumor. This suggests a dual effect of PF-562271 on inhibiting Pyk2/FAK in glioma cells—first, through the direct inhibition of Pyk2/FAK within glioma cells and second, through the reduction of TIM infiltration—consequently, attenuating the cytokine-related activation of Pyk2/FAK within glioma cells. Recurrent tumors arise from a distinct population of cells located at the invasive margin, exhibiting molecular and genetic differences, compared to primary tumor bulk. These modifications, along with alterations in the tumor microenvironment caused by the surgical resection procedure, and modulations in the TIM polarization and cytokines expression profile may contribute to the observed upregulation of Pyk2 and FAK signaling in regrown tumors. The post-surgical use of the Pyk2/FAK inhibitor PF-562271 effectively reverses Pyk2/FAK signaling activation, leading to reduced proliferation in recurrent tumors and decreased infiltration of TIMs (Figure 3 and Figure 4). 

Given the evidence of Pyk2 and FAK expression in myeloid cells [49], PF-562271 may also impair the migratory capabilities of these cells, preventing significant infiltration into the tumor resection site and reducing the TIM component involved in tumor recurrence. Nevertheless, considering the inherent heterogeneity of GBM tumors and the substantial genetic variability observed across patients, it becomes imperative to conduct further investigations using GBM animal models encompassing diverse genetic profiles. Moreover, given the previous findings highlighting PF-562271’s potential to enhance the cytotoxic effects of temozolomide, it is crucial to conduct in-depth investigations into combination therapies, primarily focused on models that represent recurrent GBM. Overall, targeting Pyk2/FAK signaling holds promise as a strategy to delay the recurrence of GBM tumors. 

## 4. Materials and Methods

All experimental procedures were carried out in accordance with the protocols approved by the Institutional Animal Care and Use Committee (IACUC, protocol # 029-2017-25-01-PHA-IBC).

### 4.1. Cell Culture 

The GL261 murine glioma cell line was obtained from the National Cancer Institute (Frederick, MD, USA). Cells were cultured in Dulbecco’s modified Eagle’s medium (DMEM) supplemented with 10% fetal calf serum (Sigma–Aldrich, St. Louis, MO, USA), 0.2 mM glutamine, 50 U/mL penicillin, and 50 μG/mL streptomycin (Sigma–Aldrich, St. Louis, MO, USA), and maintained in a humidified atmosphere of CO_2_/air (5%/95%) at 37 °C. The medium was replaced with fresh culture medium about every 3–4 days. 

### 4.2. Intracranial Glioma Implantation and Tumor Resection

C57Bl/6 mice were obtained from the Jackson Laboratory. GL261 glioma cells were implanted into the right cerebral hemisphere of 12–20-week-old C57BL/6 mice. The study included animals of both sexes. The mice were anesthetized with isoflurane, and a midline incision was made on the scalp. At stereotaxic coordinates of 2 mm lateral and 1 mm caudal to bregma, a small burr hole (0.5 mm diameter) was drilled in the skull. An aliquot (1 μL) of cell suspension (1 × 10^4^ cells/μL in PBS) was delivered at a depth of 3 mm over 1 min. 

The brain tumors were resected 14 days after implantation, as we described previously [49]. The cavity was copiously irrigated, and the skin was closed. The animals were sacrificed 14 days after resection, and the brain tissue was used for further studies. The IBox Explorer 2610 In Vivo Fluorescent Imaging System in combination with intracranial implantations of green fluorescent protein (GFP)–GL261cells and hematoxylin and eosin (H&E) staining of brain sections encompassing the tumor resection area were used to monitor the efficacy of resection, as previously demonstrated [25]. 

### 4.3. Oral Gavage

Beginning two days before the tumor resection and continuing throughout the post-surgical period, the animals received treatment through oral gavage without interruption: vehicle (100 μL of PBS) or PF-562271 (50 mg/kg daily), administered twice per day. This specific dose of PF-562271 was previously established as the optimal oral dosage for mice in effectively inhibiting FAK and Pyk2 within tumors [36,39]. No adverse effects, such as weight loss, abnormal posture, or bleeding, were observed at the provided concentration of PF-562271. 

Following the two-week treatment period with either vehicle or PF-562271, the animals were euthanized to prevent notable health deterioration stemming from tumor enlargement. This was followed by subsequent analyses involving Western blotting, immunohistochemistry, and immunofluorescence.

### 4.4. Western Blot Analysis

Cell lysates (20 μg), separated by 10% SDS–PAGE, were transferred to PVDF membranes and probed with anti-phospho-Pyk2 (Tyr 579/580) (#44636G, Thermo Fisher Scientific, Invitrogen, MA, USA), anti-Pyk2, anti-phospho-FAK (Tyr 925), anti-FAK, and anti-Cyclin D1 (#3480, #3284, #3285, #55506 Cell Signaling, Danvers, MA, USA) antibodies, followed by the addition of the corresponding secondary antibodies (cat. #A9169, Sigma–Aldrich, St. Louis, MO, USA). Detection was performed with enhanced chemiluminescence substrate (#34075, SuperSignal West Dura Extended Duration Substrate; Pierce, Rockford, IL, USA). The signal intensity was measured using a gel documentation system (Versa Doc Model 1000, Bio-Rad, Hercules, CA, USA). 

### 4.5. Tumor Size Evaluation 

Frozen 15 μm brain sections encompassing entire tumors were stained with H&E. Tumor size was calculated as the tumor area times the section thickness in all sections encompassing the tumor. 

### 4.6. Immunofluorescence Imaging 

Twenty-five micrometer frozen sections of tumors were processed with anti-Ki67 antibody (#12075, Cell Signaling, Danvers, MA, USA), followed by Taxes Red-conjugated IgG (#TI-1000, Vector Laboratories, Burlingame, CA, USA) and Lectin (Galectin, Alexa Flour 488 conjugated, #FLK-2100, Vector Laboratories, Burlingame, CA, USA). Sections were visualized using an Olympus Fluoview FV1000 confocal microscope (Olympus, Japan) and processed using ImageJ 1,54b software (http://imagej.nih.gov/ij, version 1.54b, assessed on 8 January 2023). 

### 4.7. Survival Analysis

Commencing two days prior to the tumor resection and persisting throughout the entirety of the post-surgical phase, the animals were administered either the vehicle or PF-562271, as outlined in Section 4.3. The treatment regimen was maintained until the animals reached the point of euthanasia following tumor resection. The mice were inspected daily, and those with body weight loss of 15%, decreased activity/responsiveness, and signs of neurological disorders were euthanized. The time between death of the animals and tumor resection was recorded. Comparison of survival curves was performed using the log-rank (Mantel–Cox) test. 

### 4.8. Human Pyk2 and FAK Gene Expression Analysis and Survival Probability Analysis

Clinical data and expression values pertaining to the genes PTK2B and PTK2, which are responsible for encoding the Pyk2 and FAK proteins (Glioblastoma (GBM), The Cancer Genome Atlas Program (TCGA) Provisional for Cancer Genomes, mRNA Expression z-Scores) were obtained from the cBioPortal, accessed on 26 March 2023, for cancer genomics (http://www.cbioportal.org), which contains annotated TCGA data [33,34]. The glioblastoma (TCGA, Firehose Legacy) microarray dataset, containing 619 samples, was directly downloaded from the cBioPortal on 26 March 2023. The microarray data were analyzed together with the overall survival status and gene-expression subtype and are presented as calculated survival propositions. 

Gene expression analysis of 540 glioblastoma cases from the Glioblastoma tumor RNA-Sequence dataset (TCGA-540-MAS5.0-u133a) for PTK2 (FAK) and PTK2B (Pyk2)) were obtained from the R2 Genomic Analysis Visualization Platform (https://hgserver1.amc.nl/cgi-bin/r2/main.cgi).

### 4.9. Statistical Analysis

Statistical probability was calculated using GraphPad Prism 9.0 software. One-way ANOVA was used for comparisons between groups, and statistical significance (*p* ≤ 0.05) was determined by Dunnett’s multiple comparisons test and unpaired t-tests with Welch’s correction. 

## 5. Conclusions

In a GL261-C57BL/6 glioma implantation model, tumors that regrew following surgical resection exhibited increased phosphorylation of Pyk2 and FAK, while the overall expression of Pyk2 and FAK remained relatively unchanged. Treatment with PF-562271, administered daily for two weeks starting 2 days prior to tumor resection, effectively reversed the elevated phosphorylation of Pyk2 and FAK in recurrent tumors. This treatment also resulted in reduced tumor size, decreased expression of Cyclin D1, a lower Ki67 proliferation index, and improved animal survival, compared to the non-treated group. Furthermore, PF-562271 treatment led to a decrease in the infiltration of myeloid cells into the regrown tumors.

## Figures and Tables

**Figure 1 ijms-24-13467-f001:**
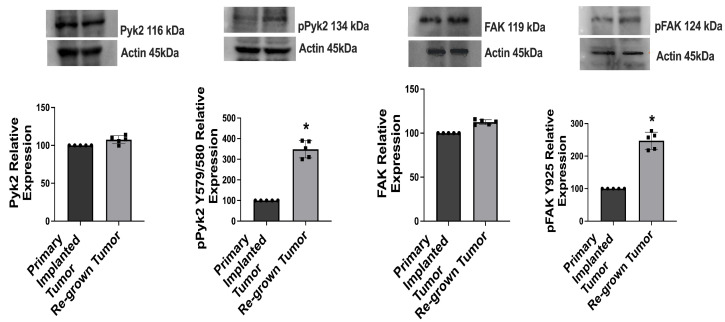
Pyk2 and FAK signaling is upregulated in tumors regrown after surgical resection, compared with primary implanted tumors in GL261-C57BL/6 mouse glioma implantation model. Representative Western blots and quantifications with individual values, for relative levels of expression of total and phosphorylated Pyk2 and FAK in primary implanted and regrown-after-resection tumors are shown. Tumors were analyzed 14 days after primary implantation (for primary implanted tumors) and 14 days after tumor resection (for regrown-after-resection tumors). Relative values in regrown tumors compared with primary implanted tumors are shown. Actin was used as a loading control. Mean ± S.E. and significant differences from the control (*) (*p* < 0.005). N = 5.

**Figure 2 ijms-24-13467-f002:**
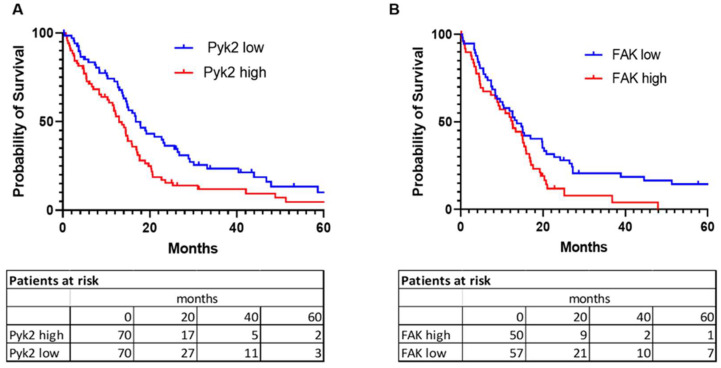
Probability of survival in glioblastoma patients depends on Pyk2 (PTK2B) and FAK (PTK2) gene expression. Kaplan–Meier survival analysis for Pyk2 (**A**) and FAK (**B**) gene expression level is shown. Microarray data analysis of 619 patients from the Cancer Genome Atlas (TCGA, Glioblastoma) were used. Curve comparison was performed with use of log-rank (Mantel-Cox) test (*p* = 0.009 for (**A**) and *p* = 0.0275 for (**B**)). Cut-off values were calculated by receiver operating characteristic (ROC) analysis.

**Figure 3 ijms-24-13467-f003:**
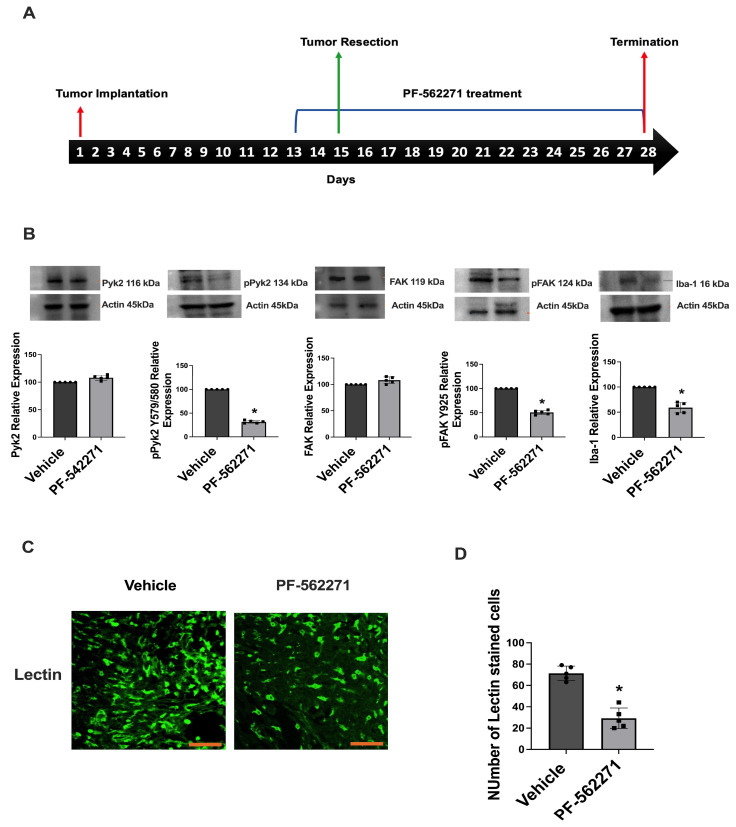
PF-562271 reduces Pyk2 and FAK phosphorylation in tumors regrown after surgical resection in a GL261-C57BL/6 mouse glioma implantation model. (**A**) Schematic representation of experimental procedures. (**B**) Representative Western blots and quantifications with individual values of relative expression of total and phosphorylated Pyk2 and FAK, together with Iba1 expression. The displayed values represent the relative differences in the PF-562271 treatment group compared to the vehicle group. (**C**) Immunofluorescence confocal images of tumors and (**D**) quantification of galactose-binding lectin-stained cells in tumors regrown after surgical resection, with and without PF-562271 treatment. PF-562271 (50 mg/kg/day) was provided orally for 14 consecutive days after resection surgery, beginning 2 days before the tumor resection procedure. Actin was used as a loading control. Scale bar is 100 µm. Mean ± S.E. and significant differences from the control (*) (*p* < 0.005). N = 5.

**Figure 4 ijms-24-13467-f004:**
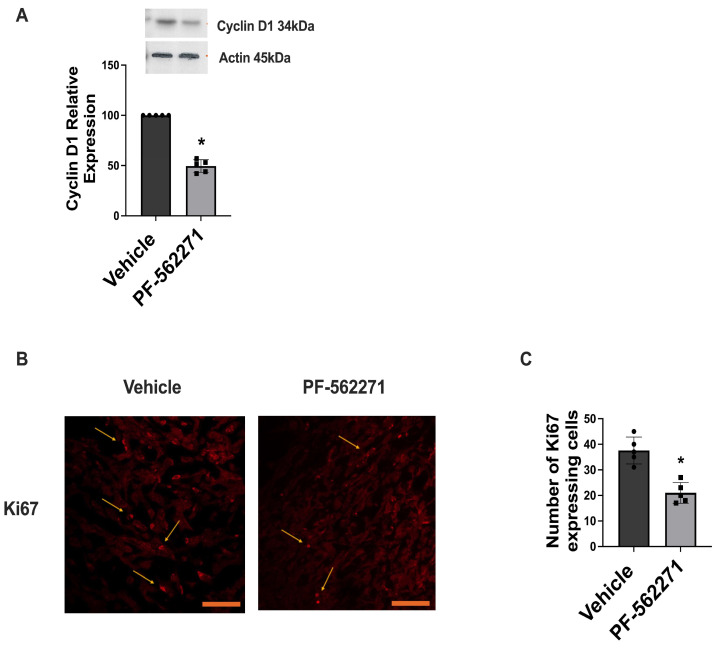
PF-562271 treatment downregulates Cyclin D1 expression and the Ki67 proliferation index in tumors regrown after surgical resection in a GL261-C57BL/6 mouse glioma implantation model. (**A**) Representative Western blots and quantifications with individual values for the relative level of Cyclin D1 protein expression in the tumors of animals that received vehicle or PF-562271 for 14 days after tumor resection. The displayed values represent the relative differences in the PF-562271 treatment group compared to the vehicle group. Actin was used as a loading control. (**B**) Representative immunofluorescence confocal images, and (**C**) a quantification of cells expressing the Ki67 marker within regrown tumors following surgical resection. The tumors were treated with PF-562271 (50 mg/kg/day) or vehicle for a continuous 14-day duration post-resection, beginning 2 days prior to the tumor removal procedure. Arrows indicate cells expressing Ki67. Scale bar is 100 µm. Mean ± S.E. and significant differences from the control (*) (*p* < 0.005). N = 5.

**Figure 5 ijms-24-13467-f005:**
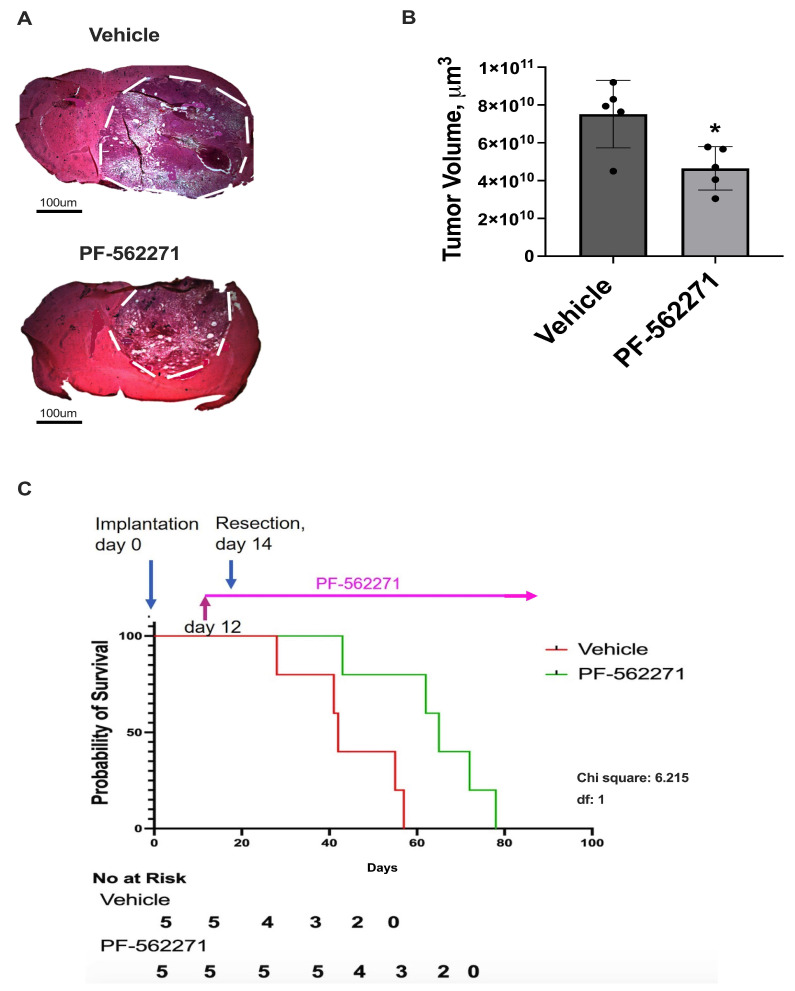
PF-562271 provided 2 days before the tumor resection and, then, during the post-surgical period reduced the growth of recurrent tumors and increased survival rates in a GL261-C57BL/6 mouse glioma implantation model. (**A**) Hematoxylin and eosin staining of brain sections, encompassing entire tumors, and (**B**) quantification with individual values of tumor size. The tumor edge is delineated by a dotted outline. The tumor resection was performed 14 days after tumor implantation. PF-562271 or vehicle were given to animals 2 days before and, then, after the tumor resection, 50 mg/kg/day, every day. Data are presented for tumors regrown 14 days after the surgical resection. Tumor size evaluation was performed immediately after the treatment course termination. (**C**) Kaplan–Meier estimates of overall survival probability for animals treated with vehicle and with PF-562271 beginning 2 days before the tumor resection and continuing during the entire post-surgical period until animal mortality. Curve comparison was performed using the log-rank (Mantel–Cox) test (*p* < 0.05). The results are presented as the mean ± S.D. with significant differences from the vehicle group (*) (*p* < 0.05). Scale bar: 100 µm. N = 5.

## Data Availability

The data generated in this study are available within the article and in the Appendix A.

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
