# Peer review of "Pyk2/FAK Signaling Is Upregulated in Recurrent Glioblastoma Tumors in a C57BL/6/GL261 Glioma Implantation Model"

_ijms, 2023, doi:10.3390/ijms241713467_

Round 1
Reviewer 1 Report
The article under consideration is devoted to the study of the role of a number of enzymes in recurrent glioblastomas. In my opinion, the article is well written, the methods used are modern and suitable for solving the problem. The conclusions are consistent with the results obtained. The practical significance of the study is undeniable. In my opinion, the article can be published as presented.
I think the study of the role of proline-reach tyrosine kinase and focal adhesion kinase in the development of glioblastoma recurrence may be of interest to specialists in this field. The adequacy of the methods used is also beyond doubt. In my opinion, the work is certainly new, original and deserves to be published in IJMS.
In terms of the quality of the presentation, I generally find it acceptable. Small remarks: it seems to me that using abbreviations in the title of an article is not a good idea. In addition, the captions for figures 3a and 3c could be more readable.
Author Response
Response to reviewers
We thank reviewers for the constructive suggestions that helped to improve the manuscript. In response to the reviewers' feedback, we addressed all reviewers’ suggestions and have made substantial revisions in the Introduction and Discussion sections, as well as rectifying and reformatting Figures 1, 3, 4, and 5. Additionally, we have incorporated the experimental treatment schedule into Figure 3 and introduced H&E staining of brain sections containing individual recurrent tumors from both the vehicle and PF-562271 treatment groups, as presented in Supplementary Figure S4.
Below we provide point by point response:
REVIEWER 1
The article under consideration is devoted to the study of the role of a number of enzymes in recurrent glioblastomas. In my opinion, the article is well written, the methods used are modern and suitable for solving the problem. The conclusions are consistent with the results obtained. The practical significance of the study is undeniable. In my opinion, the article can be published as presented.
I think the study of the role of proline-reach tyrosine kinase and focal adhesion kinase in the development of glioblastoma recurrence may be of interest to specialists in this field. The adequacy of the methods used is also beyond doubt. In my opinion, the work is certainly new, original and deserves to be published in IJMS.
In terms of the quality of the presentation, I generally find it acceptable. Small remarks: it seems to me that using abbreviations in the title of an article is not a good idea. In addition, the captions for figures 3a and 3c could be more readable.
Since the abbreviation "Pyk2" for "proline-rich tyrosine kinase 2" might not be immediately clear, we have retained the "Pyk2/FAK" abbreviation in the title to prevent any potential confusion. Additionally, we have made adjustments to the caption and font size in Figure 3.
Reviewer 2 Report
The authors showed that FAK/Ryk2 signaling is upregulated in recurrent GBMs. They reported that inhibition of FAK/Ryk2 activities reduce the signaling activities and reduced tumor volume. It may be an interesting discovery however, more information is needed.
The authors reported the similar results in their recent study appeared in J Neuro-oncol 161, 593–604 (2023). https://doi.org/10.1007/s11060-023-04260-3. Although the authors showed the different insights, as a reader, it is not easy to find a merit of this study. The manuscript is well written, but it will be better if the authors add more experimental results such as the effect of TMZ and TMZ+PF-562271 in GBM.
Author Response
Response to reviewers
We thank reviewers for the constructive suggestions that helped to improve the manuscript. In response to the reviewers' feedback, we addressed all reviewers’ suggestions and have made substantial revisions in the Introduction and Discussion sections, as well as rectifying and reformatting Figures 1, 3, 4, and 5. Additionally, we have incorporated the experimental treatment schedule into Figure 3 and introduced H&E staining of brain sections containing individual recurrent tumors from both the vehicle and PF-562271 treatment groups, as presented in Supplementary Figure S4.
Below we provide point by point response:
REVIEWER 2
The authors showed that FAK/Ryk2 signaling is upregulated in recurrent GBMs. They reported that inhibition of FAK/Ryk2 activities reduce the signaling activities and reduced tumor volume. It may be an interesting discovery however, more information is needed.
The authors reported the similar results in their recent study appeared in J Neuro-oncol 161, 593–604 (2023). https://doi.org/10.1007/s11060-023-04260-3. Although the authors showed the different insights, as a reader, it is not easy to find a merit of this study. The manuscript is well written, but it will be better if the authors add more experimental results such as the effect of TMZ and TMZ+PF-562271 in GBM.
In prior investigations, we highlighted the increased Pyk2 and FAK signaling in GBMs upon exposure to TMZ treatment and the release of soluble factors from microglia within the tumor environment. Building on these earlier findings, we propose that the observed elevation of Pyk2 and FAK in regrown GBMs following surgical removal stems from sustained alterations in cytokine expression by tumor-infiltrating microglia, a phenomenon we previously documented, coupled with distinctive characteristics of GBM cells at the invasive margin. These specific cells act as a source for the regenerated tumor.
A more comprehensive exploration of the mechanisms driving the heightened Pyk2 and FAK activity in recurrent tumors is necessary and will be explored in separate publications. In this present report, we emphasize that the heightened Pyk2/FAK signaling in GBM recurrence post-surgical resection contributes to an aggressive tumor growth trajectory. This factor should be considered when formulating treatment strategies for recurrent GBMs.
Within the scope of this study, we did not delve into the effects of TMZ or TMZ+PF-562271. This choice was mainly influenced by the infrequent utilization of TMZ for recurrent GBMs, with targeted therapies being the primary approach. Additionally, considering our recent publication in Journal of Neuro-Oncology on the combined TMZ+PF-562271 regimen for both murine models with primary implanted GBMs and primary human GBMs, repetitive presentation of these findings in the current manuscript might be redundant.
Reviewer 3 Report
The related manuscript discusses the role of phosphorylated Pyk2 and FAK expressions on cell cycle and invasion regulation in primary implanted tumor and re-grown glioblastoma tissues. In addition, the effect of the Pyk2/FAK inhibitor PF-562271 on survival rate was investigated. The quality of the manuscript is generally good, but there are some remarkable questions here. These are;
11. In the introduction part of the manual, it was stated that Pyk2 and FAK molecules are involved in the cell cycle and invasion regulation in glioblastoma, depending on their high expression levels. However, in the results section, there was no significant difference in nonphosphorylated states of Pyk2 and FAK expression in primary implanted tumor and re-grown tumor glioblastoma tissues, while a significant increase in the expression of Pyk2 and FAK phosphorylated states was determined. This situation raises questions about the mechanism of what is stated in the introduction. For this reason, the relevant mechanism is expected to be clarified in the introduction.
22. The font sizes of the explanations in the figure 1 and 3 are small. For the explanations to be read better, larger writing fonts will increase the readability of the figure.
33. Line 159: The abbreviation "Iba1" for "Ionized Calcium-Binding Adapter Molecule 1" should be given here. Please check whether this is the case for other abbreviations. Also, why is the relationship between Iba1 expression and phosphorylated Pyk2 and FAK expression levels important? This situation should be noted here or introduction section.
Author Response
Response to reviewers
We thank reviewers for the constructive suggestions that helped to improve the manuscript. In response to the reviewers' feedback, we addressed all reviewers’ suggestions and have made substantial revisions in the Introduction and Discussion sections, as well as rectifying and reformatting Figures 1, 3, 4, and 5. Additionally, we have incorporated the experimental treatment schedule into Figure 3 and introduced H&E staining of brain sections containing individual recurrent tumors from both the vehicle and PF-562271 treatment groups, as presented in Supplementary Figure S4.
Below we provide point by point response:
REVIEWER 3
The related manuscript discusses the role of phosphorylated Pyk2 and FAK expressions on cell cycle and invasion regulation in primary implanted tumor and re-grown glioblastoma tissues. In addition, the effect of the Pyk2/FAK inhibitor PF-562271 on survival rate was investigated. The quality of the manuscript is generally good, but there are some remarkable questions here. These are;
In the introduction part of the manual, it was stated that Pyk2 and FAK molecules are involved in the cell cycle and invasion regulation in glioblastoma, depending on their high expression levels. However, in the results section, there was no significant difference in nonphosphorylated states of Pyk2 and FAK expression in primary implanted tumor and re-grown tumor glioblastoma tissues, while a significant increase in the expression of Pyk2 and FAK phosphorylated states was determined. This situation raises questions about the mechanism of what is stated in the introduction. For this reason, the relevant mechanism is expected to be clarified in the introduction.
Given their involvement in intracellular regulatory processes, the extent of phosphorylation in Pyk2 and FAK isn't directly correlated with the overall expression levels of these kinases. In this manuscript, we emphasize the activation (phosphorylation) of Pyk2 and FAK signaling in regrown tumors, rather than shifts in their total expression. We have made revisions to the sentence in the Introduction section (lines 90-91) to address this potential confusion.
The font sizes of the explanations in the figure 1 and 3 are small. For the explanations to be read better, larger writing fonts will increase the readability of the figure.
In the revised manuscript, we have increased the font size in the specified figures.
Line 159: The abbreviation "Iba1" for "Ionized Calcium-Binding Adapter Molecule 1" should be given here.
We made revisions to the sentence in lines 190-191 to include abbreviation for Iba-1.
Please check whether this is the case for other abbreviations. Also, why is the relationship between Iba1 expression and phosphorylated Pyk2 and FAK expression levels important? This situation should be noted here or introduction section.
We have now provided a rationale for employing the Iba1 marker in both PF-562271 treated and control animals in lines 188-192. Notably, given the established Pyk2/FAK dependence of microglia/macrophages according to Avraham et al. (2000), the inhibition of these kinases can potentially influence their tumor infiltration, consequently resulting in alterations within the tumor microenvironment.
Reviewer 4 Report
The manuscript by Ortiz Rivera et al. used the orthotopic murine glioma model GL261 in C57BL/6 to analyze the effect of Pyk2/FAK inhibitors on tumor regrowth and protein expression following resection. The authors analyzed the protein expression of Pyk2 and FAK via Western blotting and compared the data to publicly available gene expression data from GBM patients. They transplanted and surgically removed GBMs from mouse brains and applied the Pyk2/FAK inhibitor PF-562271 over several days before and after the surgery. The (re)-grown tumors were analyzed for several markers (Cyclin D1, Ki67 & others) by Western blotting and immunofluorescence stainings.
There is a high medical need for new and innovative treatment approaches for GBM patients. The manuscripts test a surgical approach in combination with an inhibitor application. More details on the statistical planning of the animal experiments are needed. It is not clear how many animals were used per endpoint. Also, please include a schedule for the different interventions and endpoints. Side effects of the treatment and tumor burden should be mentioned. How was tumor growth monitored?
Specific comments:
Abstract
line 12: typo, please change to proline-rich
Introduction:
Line 32: GBMs recur shortly after resection – please be more precise here. Provide a time span in e.g. months.
Line 41-44: is this change in profile due to genetics or also seen on protein level?
Line 104-105: What is the rational for this application schedule? Why was the experiment terminated after 2 weeks? Add that info in M&M.
Results:
The relevance of phosphorylated Pyk2 and FAK is not introduced. Please add.
Please repeat in 2.2. the gene names for Pyk2 and FAK. In the text are the gene names, in the figure legend the protein names. How is the survival of double low (Pyk2 & FAK) / double high/ mixed patients?
Line 155: analysis was performed on 14th day of treatment. Should that not be day 16? Maybe adding a time line in M&M would be advantageous. D1 injection; D12 first treatment day, D14 resection, D28 termination …Line 112-114 reads 14 days after implantation and 14 after resection.
Line 159: Iba1 is not introduced and not discussed.
Line 161: TIM is not introduced.
Figure 3C is too small
Figure 5: Histological sections should be analyzed by a GBM pathologist. The Tumor growth is massive. Please provide a scale bare! Although, the magnification is low, bleeding, necrosis, and brain compression are obvious. Which side effects were detected? Panel C) please change the labeling of the x axis in e.g. days.
The experiment was performed with five animals per group? Please provide statistical planning for the experiment. What is the rational for 50 mg/kg daily? A curve comparison was performed (Fig 5C, log-rank (Mantel‒Cox))? Please add the result in the figure.
Please provide an overview of the used animals for the other experimental parts.
Discussion
Line 260 ff: Treatment-induced pH value shifts were discussed as activators for Pyk2. Did you measure if the application of the inhibitor shifts the pH value as well?
In general, the authors speculate on the invasive potential, migratory potential, and proliferation of GBM cells. Did you test these endpoints in vitro using your cell model system and the inhibitor?
The limitation of your study needs to be discussed eg. Your approach comprising surgery + inhibitor versus clinical standard incl surgery, radiotherapy, and TMZ.
Material and Methods:
Line 319: please add that GL261 are murine cells
Line 337: what is the sex of the used animals?
Line 340: 1 x104 cells
Line 342: The tumors were resected 14 days following implantation. Please add a reference or more details on how this was done. How did you measure tumor size in vivo?
Conclusion:
Line 395: improved animal survival
Minor editing of English language required
Author Response
Response to reviewers
We thank reviewers for the constructive suggestions that helped to improve the manuscript. In response to the reviewers' feedback, we addressed all reviewers’ suggestions and have made substantial revisions in the Introduction and Discussion sections, as well as rectifying and reformatting Figures 1, 3, 4, and 5. Additionally, we have incorporated the experimental treatment schedule into Figure 3 and introduced H&E staining of brain sections containing individual recurrent tumors from both the vehicle and PF-562271 treatment groups, as presented in Supplementary Figure S4.
Below we provide point by point response:
REVIEWER 4
The manuscript by Ortiz Rivera et al. used the orthotopic murine glioma model GL261 in C57BL/6 to analyze the effect of Pyk2/FAK inhibitors on tumor regrowth and protein expression following resection. The authors analyzed the protein expression of Pyk2 and FAK via Western blotting and compared the data to publicly available gene expression data from GBM patients. They transplanted and surgically removed GBMs from mouse brains and applied the Pyk2/FAK inhibitor PF-562271 over several days before and after the surgery. The (re)-grown tumors were analyzed for several markers (Cyclin D1, Ki67 & others) by Western blotting and immunofluorescence stainings.
There is a high medical need for new and innovative treatment approaches for GBM patients. The manuscripts test a surgical approach in combination with an inhibitor application. More details on the statistical planning of the animal experiments are needed. It is not clear how many animals were used per endpoint. Also, please include a schedule for the different interventions and endpoints. Side effects of the treatment and tumor burden should be mentioned. How was tumor growth monitored?
In section 4.3 of the Methodology, we have introduced a description of the potential treatment side effects and the duration of treatment. Our extensive experience with mouse glioma implantation models over a 3-week period post-implantation has not indicated significant side effects resulting from tumor burden. This observation holds true until the tumor size considerably affects the health of the mice. Given that tumor resection and PF-562271 treatments were carried out within 2 weeks of tumor implantation, we did not mention any side effects associated with tumor burden.
Furthermore, drawing from our familiarity with mouse glioma implantation models, the variability in tumor growth and positioning among individual animals remains under 15%. We have evaluated tumor development dynamics and positions across several of our prior studies using H&E staining (Ortiz et al., 2022, 2023; Rolon-Reyes et al., 2015), which has enabled us to establish the optimal tumor size for the resection procedure as two weeks post-implantation.
We have included the number of animals used in lines 182 and 283. The schedule of experimental procedures has been integrated into Figure 3A.
Specific comments:
Abstract
line 12: typo, please change to proline-rich
We corrected the typo in abstract section.
Introduction:
Line 32: GBMs recur shortly after resection – please be more precise here. Provide a time span in e.g. months.
We have now provided the average duration, measured in months, for the recurrence of GBM.
Line 41-44: is this change in profile due to genetics or also seen on protein level?
The referenced literature elucidated alterations in gene expression profiles between primary tumors and those that re-grow. In order to prevent any potential misunderstanding, we have included precise clarifications within this sentence.
Line 104-105: What is the rational for this application schedule? Why was the experiment terminated after 2 weeks? Add that info in M&M.
The decision to implement a 14-day treatment schedule was driven by the intention to prevent substantial health decline associated with tumor regrowth. However, this 14-day treatment constraint did not extend to survival analysis, where treatment continued until the animals reached the euthanasia point. These explanations have now been incorporated into sections 4.3 and 4.7.
Results:
The relevance of phosphorylated Pyk2 and FAK is not introduced. Please add.
The significance of phosphorylated Pyk2 and FAK was now added in the Introduction, specifically in lines 85 to 97.
Please repeat in 2.2. the gene names for Pyk2 and FAK. In the text are the gene names, in the figure legend the protein names.
We now incorporated gene names in line 158 and in Fig.2 figure legend.
How is the survival of double low (Pyk2 & FAK) / double high/ mixed patients?
We attempted an analysis using different tiers of both Pyk2 and FAK gene expressions within GBM RNA-Seq datasets sourced from various platforms (TCGA, cBio, R2: Genomic Analysis and Visualization Platform). Unfortunately, none of the datasets provide a sufficient quantity of RNA-sequenced specimens that encompass a wide range of Pyk2 and FAK gene expression levels, necessary for robust statistical analysis. Larger RNA-Seq datasets are essential for conducting this form of analysis. As a result, we chose not to include this analysis in the manuscript.
Line 155: analysis was performed on 14th day of treatment. Should that not be day 16? Maybe adding a time line in M&M would be advantageous. D1 injection; D12 first treatment day, D14 resection, D28 termination …Line 112-114 reads 14 days after implantation and 14 after resection.
Enhanced elucidation of the treatment procedures has been included in several sections: lines 132-134, 280-282, and 292-293 in Results section, as well as sections 4.3 and 4.7 within the methodology section. Furthermore, a visual depiction of the experimental procedures has been integrated into Fig. 3A.
Line 159: Iba1 is not introduced and not discussed.
We have added a justification for utilizing the Iba1 marker in both PF-562271 treated and control animals in lines 187-193. Furthermore, we have incorporated a discussion on the modulation of the Iba1 marker due to PF-562271 treatment, as outlined in lines 393-401.
Line 161: TIM is not introduced.
The term "TIM" is now introduced in line 95 of the Introduction.
Figure 3C is too small
Adjustments to the font size in Figure 3 have been implemented.
Figure 5: Histological sections should be analyzed by a GBM pathologist. The Tumor growth is massive. Please provide a scale bare! Although, the magnification is low, bleeding, necrosis, and brain compression are obvious. Which side effects were detected? Panel C) please change the labeling of the x axis in e.g. days.
In a previous study, we documented that PF-562271 monotherapy yielded a non-significant increase in apoptosis and a substantial reduction in the invasion margin within tumors in the GL261/C57Bl/6 model (Ortiz et al., 2023). Consequently, to prevent redundancy, we opted not to include a detailed account of these aspects in the manuscript. Additionally, we observed no significant distinctions in the occurrence of necrosis or bleeding within tumors, and no considerable side effects linked to PF-562271 treatment. These observations have been now added in lines 285-287.
Furthermore, we have integrated a scale bar into Figure 5A and adjusted the labeling on the x-axis in Figure 5C.
The experiment was performed with five animals per group?
Five animals per group were employed. This information is now emphasized in lines 183 and 283 of the Results section, alongside the figure legends.
Please provide statistical planning for the experiment. What is the rational for 50 mg/kg daily? A curve comparison was performed (Fig 5C, log-rank (Mantel‒Cox))? Please add the result in the figure.
We have introduced appropriate references to provide the rationale for the PF-562271 dosage in the mouse GBM model, which can be found in lines 450-451. Additionally, statistical results for the comparison of the curves in Figure 5C have been incorporated.
Please provide an overview of the used animals for the other experimental parts.
The number of animals used is now emphasized in lines 183 and 283 of the Results section, alongside the figure legends.
Discussion
Line 260 ff: Treatment-induced pH value shifts were discussed as activators for Pyk2. Did you measure if the application of the inhibitor shifts the pH value as well?
Although we mentioned pH values as potential factors for Pyk2 activation in GBM tumors, we chose not to delve further into investigating this effect, as it falls outside the scope of our current research focus.
In general, the authors speculate on the invasive potential, migratory potential, and proliferation of GBM cells. Did you test these endpoints in vitro using your cell model system and the inhibitor?
In a previously conducted study within our laboratory, we employed GBM mouse models and primary human cell lines to explore the impacts of PF-562271 on GBM proliferation, migration, and invasion capabilities. This information has been integrated into lines 325-332 of the Discussion section.
The limitation of your study needs to be discussed eg. Your approach comprising surgery + inhibitor versus clinical standard incl surgery, radiotherapy, and TMZ.
The model's limitations and future research directions are now emphasized in lines 411-416.
Material and Methods:
Line 319: please add that GL261 are murine cells
This correction has been implemented.
Line 337: what is the sex of the used animals?
In our prior investigations utilizing the GL261/C57Bl/6 model, no substantial molecular or physiological distinctions were observed in GBM tumors between the sexes. Consequently, the study included both female and male animals; however, the results were individually analyzed for each gender. Since no significant differences were identified, the data were consolidated. This clarification has been incorporated into section 4.2.
Line 340: 1 x104 cells
This correction was implemented.
Line 342: The tumors were resected 14 days following implantation. Please add a reference or more details on how this was done. How did you measure tumor size in vivo?
We have now included a reference to our previously published protocol.
Conclusion:
Line 395: improved animal survival
The correction was implemented.
Reviewer 5 Report
The work done by Rivera et. al. addresses an important aspect in the field of recurring glioblastoma therapy however the manuscript needs extensive revision before it's acceptance. Hereby, I list a few comments that will help in making the manuscript more interesting and informative.
1. The information in line numbers 48 to 70 should be briefly stated in just 2 or 3 sentences, if at all necessary. To me I don't see this information to be of much relevance pertaining to the study and is making the introduction unnecessarily lengthy.
2. In figure 1, why are there no error bars shown for the bar diagrams showing the quantification of molecular expression in primary implanted tumors? Also, the western blots were repeated how many times? please include individual data value points from each blot in the bar graphs.
3. In figure 3 A, how can all the beta actin blots provided be identical? Were all the molecules shown here, run, and developed from the same gel? If yes, this isn't the right way to represent the data. The whole blot should be provided and the band showing specific/ representative molecule should be labeled based on their molecular weight. Please reconstruct the figure based on the suggestion.
4. In figure 3 B, can I see zoomed images of a few lectin positive cells in both vehicle and control for identifying closer morphological changes between the cells in 2 groups, if any?
5. Please provide blots with original backgrounds without any editing. Here the authors have increased the brightness and contrast of the western blot images so much that all background is gone making the blots look unscientific. Further, western blots should never be cropped so much. This mostly applies to all the western blot images and especially the Figure 4 A.
6. Please provide quantification for figure 4C.
7. A key experiment missing in the manuscript is tumor histology after resection in both the conditions, to make sure that the parameters were similar during the recurrence and to identify the distinct cell population in the invasive margins.
8. Once again, the image provided in figure 5 A isn't a proper scientific way of representation. Nothing much can be understood from the image. Please provide bigger add clearer images with higher magnification. Also, label the important regions for better visualization of the differences between the 2 conditions.
9. Include images of the individual recurrent tumors, exclusively, for both the vehicle and inhibitor groups in order to compare their size between the groups.
10. Like the introduction, the discussion is also way too long. Please rewrite making it brief, informative and interesting to the readers.
Author Response
Response to reviewers
We thank reviewers for the constructive suggestions that helped to improve the manuscript. In response to the reviewers' feedback, we addressed all reviewers’ suggestions and have made substantial revisions in the Introduction and Discussion sections, as well as rectifying and reformatting Figures 1, 3, 4, and 5. Additionally, we have incorporated the experimental treatment schedule into Figure 3 and introduced H&E staining of brain sections containing individual recurrent tumors from both the vehicle and PF-562271 treatment groups, as presented in Supplementary Figure S4.
Below we provide point by point response:
REVIEWER 5
The work done by Rivera et. al. addresses an important aspect in the field of recurring glioblastoma therapy however the manuscript needs extensive revision before it's acceptance. Hereby, I list a few comments that will help in making the manuscript more interesting and informative.
- The information in line numbers 48 to 70 should be briefly stated in just 2 or 3 sentences, if at all necessary. To me I don't see this information to be of much relevance pertaining to the study and is making the introduction unnecessarily lengthy.
We have now condensed this section of the Introduction.
- In figure 1, why are there no error bars shown for the bar diagrams showing the quantification of molecular expression in primary implanted tumors? Also, the western blots were repeated how many times? please include individual data value points from each blot in the bar graphs.
We have illustrated the relative values in the experimental group in comparison to the control group. As a result, all control groups have been normalized to 100%. This clarification has been included in the figure legends for Figures 1, 3, and 4.
A total of five animals per group were analyzed. We have incorporated these clarifications in lines 183 and 283, as well as in all the figure legends.
The Western blot graphs were adjusted to display individual data value points.
- In figure 3 A, how can all the beta actin blots provided be identical? Were all the molecules shown here, run, and developed from the same gel? If yes, this isn't the right way to represent the data. The whole blot should be provided and the band showing specific/ representative molecule should be labeled based on their molecular weight. Please reconstruct the figure based on the suggestion.
We have corrected this error in Figure 1. Complete Western blot images along with corresponding loading controls are provided in Supplementary Figures S2 and S3.
- In figure 3 B, can I see zoomed images of a few lectin positive cells in both vehicle and control for identifying closer morphological changes between the cells in 2 groups, if any?
Above, we have provided two enlarged images for each vehicle and PF-562271 treatment, taken from distinct regions within the entire images. It's important to note that lectin might not be the most precise marker for characterizing cell morphology. For a more accurate analysis, actin staining, focal adhesion markers, and invadopodia formation assays would be more suitable. Nevertheless, our study's primary aim was to assess myeloid cell infiltration levels within tumors, rather than examining the impact of PF-562271 on TIM functions and phenotype, which necessitates an in-depth individual investigation. While lectin staining could provide an initial insight into the more ramified TIM cell phenotype, we refrain from discussing this aspect in the manuscript due to the reasons mentioned.
- Please provide blots with original backgrounds without any editing. Here the authors have increased the brightness and contrast of the western blot images so much that all background is gone making the blots look unscientific. Further, western blots should never be cropped so much. This mostly applies to all the western blot images and especially the Figure 4 A.
We made corrections to western blot images. Additionally, uncropped unmodified images are provided in supplementary Figures S2 and S3.
- Please provide quantification for figure 4C.
We reorganized Figure 4, positioning panels B and C for improved visual representation.
- A key experiment missing in the manuscript is tumor histology after resection in both the conditions, to make sure that the parameters were similar during the recurrence and to identify the distinct cell population in the invasive margins.
The tumor resection protocol, along with the strategies employed to assess resection effectiveness such as the In Vivo Fluorescent Imaging System and hematoxylin & eosin (H&E) staining of brain sections encompassing the resected tumor area, were detailed in a previous publication by Ortiz-Rivera (2022). To avoid duplicating data already presented in the previous publication, we have included a reference in the Methodology section (4.2) to substantiate the protocol instead of presenting images illustrating resection effectiveness and tumor regrowth dynamics. This approach is taken due to the consistently low variability observed in the growth and positioning of implanted tumors, thereby ensuring a consistent assessment of resection efficacy.
- Once again, the image provided in figure 5 A isn't a proper scientific way of representation. Nothing much can be understood from the image. Please provide bigger add clearer images with higher magnification. Also, label the important regions for better visualization of the differences between the 2 conditions.
We have increased the size of the H&E images in Figure 5A and accentuated the tumor edges for enhanced visualization. However, due to limitations in accommodating the entire brain section within the field of view, we are unable to provide images at a higher magnification.
- Include images of the individual recurrent tumors, exclusively, for both the vehicle and inhibitor groups in order to compare their size between the groups.
H&E staining for brain sections encompassing individual recurrent tumors from both the vehicle and PF-562271 treatment groups are now presented in Supplementary Figure S4.
- Like the introduction, the discussion is also way too long. Please rewrite making it brief, informative and interesting to the readers.
We have revised the discussion section, retaining only the most pertinent information aligned with the scope of our study.
Round 2
Reviewer 5 Report
The authors stated in their reply that they have adjusted the bar graphs in Figure 1 to display individual data value points, however the images provided don't reflect such changes as 5 individual data points are missing. Same thing for figure 3B and 4A. Please provide individual points in bar graphs like figure 5B.
Similarly, the authors answered "We reorganized Figure 4, positioning panels B and C for improved visual representation." for Figure C but they haven't provided the quantification with individual values. Revise and present as requested.
Author Response
We thank the reviewer for the constructive suggestions that helped improve ourmanuscript. We addressed the reviewersuggestions and performed pertinent modifications for Figure 1,3 and 4.
Below we provide point by point response:
The authors stated in their reply that they have adjusted the bar graphs in Figure 1 to display individual data value points, however the images provided don't reflect such changes as 5 individual data points are missing. Same thing for figure 3B and 4A. Please provide individual points in bar graphs like figure 5B.
Similarly, the authors answered "We reorganized Figure 4, positioning panels B and C for improved visual representation." for Figure C but they haven't provided the quantification with individual values. Revise and present as requested.
We now made all the corresponding and appropriate modification in Figures 1,3 and 4 as requested by the reviewer. Each graph is displayed as individual value points for figure 1,3 and 4.